# Performance Comparison of the Vertical and Horizontal Oriented Stirred Mill: Pilot Scale IsaMill vs. Full-Scale HIGMill

**Metin Can *** and **Okay Altun**

Department of Mining Engineering, Hacettepe University, Beytepe, Ankara 06800, Turkey
* Correspondence: metin.can@hacettepe.edu.tr; Tel.: +90-3122977600; Fax: +90-3122992155

**Abstract:** Varied types/geometries of stirred mills have been produced by different manufacturers, and the comparison task has been accomplished for some of the technologies, i.e., Tower mill vs IsaMill. However, the main drawbacks of these comparisons were the uncommon characteristics of the milling environment, such as media size. In this study, HIGMill and IsaMill, which were vertically and horizontally chamber oriented, respectively, were compared for a regrinding process of copper ores with similar characterization and almost the same milling environment. Detailed characterization studies of the two ore types, such as work index, ore breakage and chemical composition, were performed. Modeling of the two mills was also performed to show the variation in the rate of breakage parameters. The entire assessments were based on comparing the signature plots, energy and shape of the product size distribution as well as the stress analyses. The results showed that HIGMill and IsaMill technologies behaved in a different manner for coarse and fine tail of comminution. IsaMill with horizontal orientation was found to be more energy-efficient, particularly at the fine grind size, and produced finer product when it was operated at the same stress level of HIGMill.

**Keywords:** stirred mill; HIGMill; IsaMill; regrind mill; comminution; stress analyses

## 1. Introduction

Stirred media mills are indispensable members of the mineral processing flowsheets, which are requiring regrind and fine grinding applications. Such a high utilization of these technologies comes from their energy-efficient operation over the conventional techniques for below the size range of 100 micron [1–4]. In addition to the energy efficiency, improvements in both grade and recovery of the concentration processes had also been reported [5]. Wills and Finch [6] deduced that the number of installations of these mills is growing steadily; hence, it is believed that the technology will keep its importance in the future.

There exist different sizes and brands which are in the serve of the minerals industry [6–8]. The well-known ones are Vertimill, SMD, VXP, HIGMill and IsaMill. Vertimill is manufactured by Metso-Outotech company and has a vertical oriented chamber. Their operational range is given as 400–30 micron [9,10], and energy saving relative to ball milling application is achievable. SMD has been in use since 1998 for the production of <15 μm particles. The mill also has vertical arrangement. VXP Mill is a product of FLSmidth company and is vertically oriented stirred media mill. The mill is often used in flotation concentrate regrind and precious metals tailings retreatment where the feed size is typically 200 μm [11,12]. HIG Mill is a recently developed technology by Metso-Outotech company. It has a vertical chamber and is used in producing submicron particles [13]. IsaMill is the only horizontal oriented mill among the mentioned brands, and its successful operations have been reported [14–17].

Stirred mills can be categorized according to their operating speed, shape of the agitator and the chamber orientation [6,17]. When the speed is considered, low-speed mills are said to be adequate for fine grinding of relatively coarser particles as it is operated

without fluidizing the beads inside. High-speed operation, on the other hand, agitates the beads vigorously; hence, the media is fluidized, and the number of collisions is improved. Such operational characteristics are better suited for ultrafine range [6,7]. So far, varied shapes of agitators have been provided, named as pins, discs or a spiral screw fixed on a main shaft to transfer energy to the mill charge. Regarding the chamber orientation, the stirred mills can either be vertical or horizontal aligned [2,3].

It has been always a debate whether to choose horizontal or vertical orientation. The main issue of the vertical design is the settling of the beads to the bottom of the chamber after a shut-down. This results in operational issues for the start-up; hence, the mechanical design of this arrangement is dominated by start-up torque [14]. Consequently, scale-up problems are encountered. On the other hand, within the horizontal configuration, many stirrers are active on fluidizing the settled load. As a result, the scale-up of the mill is said to be relatively easier. There have been some attempts focusing on the performance comparison of the two arrangements. Previously, the comminution results of Tower Mill and IsaMill were compared [3,14,15]. Since IsaMills are operated at higher tip speed with lower media size, it was named as energy-efficient technology over the vertical arrangement. Another performance parameter is the shape of the product size distributions. It has been shown that the Tower Mill produces product with wider size distribution compared to IsaMill operated at the same target size [14]. However, when the operational ranges of the two technologies are considered, it is realized that the feed and product sizes are noticeably different [6]. The Tower Mills can cope with 6 mm top size; hence, it should utilize coarser balls inside the chamber. Therefore, the comparisons of the two mills may not be appropriate for fine grinding range. The study held by Parry [18] compared SMD technology and horizontal stirred media mill, called Netzsch mill. The test works completed at different streams of the flotation circuit summarized that SMD technology produced the target size with less or the same energy utilization to that of the Netzsch mill. The difference was more obvious at fine size production <8μm. A recent study investigated the influences of the chamber orientation on the grinding results for the same mill [19]. Within the study, the comparison was made on dry batch grinding of calcite at the same milling environment. Assessments of the product size deduced that horizontal configuration generated more fines. Moreover, the stress analyses pointed out that horizontal configuration was more energy efficient.

Within the well-known technologies, HIGMill and IsaMill have common milling environments, i.e., tip speed, mill filling, bead size, etc., as well as similar feed and product size ranges [6,7,13]. Hence, comparison of these two mills is expected to give an impression of the energy efficiencies of the orientations that has not been studied so far. The novelty of the study is to fulfill this requirement by giving some insight into the performance variations. Within the scope of the study, the performance of industrial scale HIGMill operation in a copper regrind circuit was compared with the results of pilot scale IsaMill of another mining operation employed for regrind purpose. Although these mills were operated by different mining companies, it is known that the ore types processed have very close geological characterizations in terms of their location, and the comparison was made based on this fact. The entire assessments were based on comparing the signature plots, stress analyses as well as the shape of product size distributions.

## 2. Materials and Methods

### 2.1. HIGMill and IsaMill

HIG mill at the site (Figure 1) has the following specifications (Table 1).

In brief, the HIGMill is bottom-fed, and the slurry is pumped via bottom connection. As the feed flows to the upper zones, the particles are subjected to a series of grinding actions imparted by the beads agitated vigorously. The product leaves the chamber at the top, and no screen or any structure exists to separate the beads since gravity keeps them within the chamber [13].

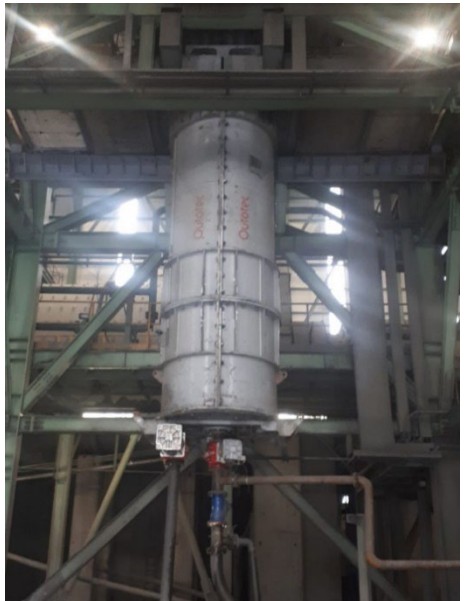
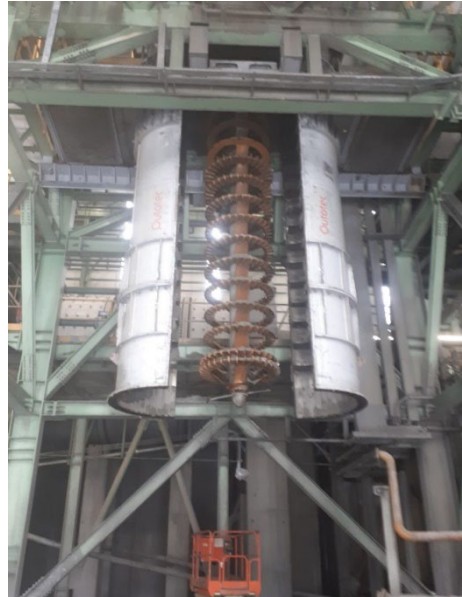

**Figure 1.** HIG mill at the copper mine.

**Table 1.** Mill specifications.

| | |
|---|---|
| Inner height (m) | 6.8 |
| Inner diameter (m) | 1.66 |
| Dia. of agitator (m) | 1.36 |
| No. of agitator | 16 |
| Net volume ($m^3$) | 13 |
| Max. available tip speed (rpm; m/s) | 1000; 10.4 |
| Installed mill motor power (kW) | 2650 |

Figure 2 illustrates the flowsheet of copper milling and beneficiation circuit in which HIGMill is used for the regrind application. The mill processes cyclone underflow material, which is diluted with the water to adjust the milling environment. The product of the mill is then sent to the cleaning stage.

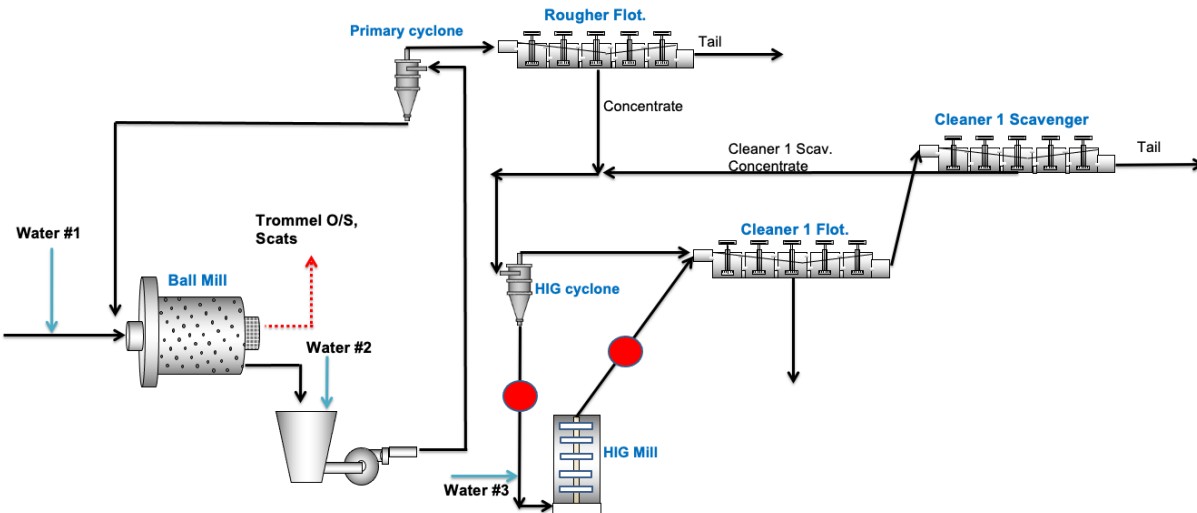

**Figure 2.** Flowsheet of milling & beneficiation circuit of the copper mine with HIGMill.

The IsaMill tests were performed at another copper mine, which has the flowsheet depicted in Figure 3. As indicated on the illustration, the mill aimed to process the cyclone underflow stream to replace the ball milling application.

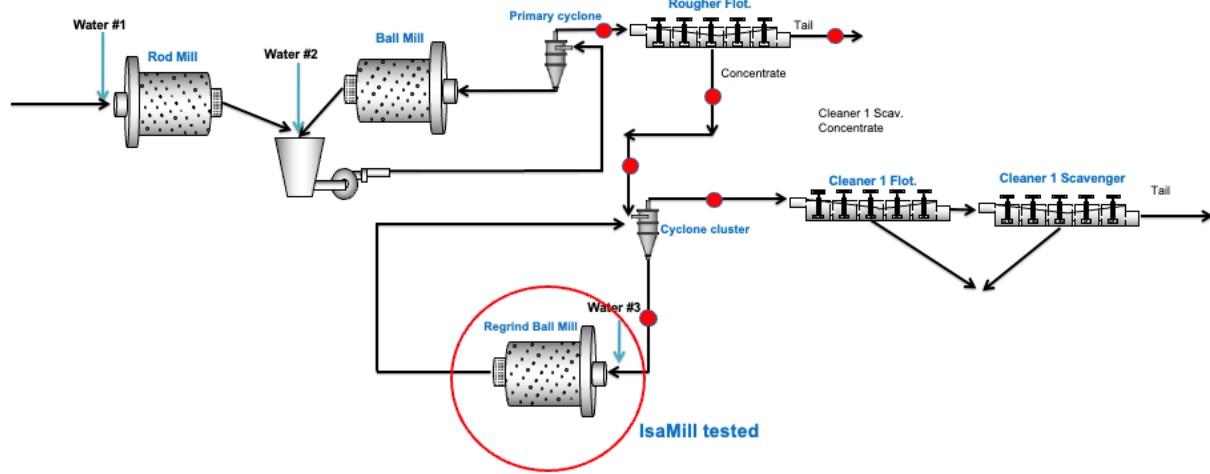

**Figure 3.** Flowsheet of milling and beneficiation circuit of the copper mine with IsaMill.

The setup of the pilot scale IsaMill M20 is illustrated in Figure 4. The process enables multi-stage grinding/passing of the feed sample; hence, the signature plot can be developed accordingly.

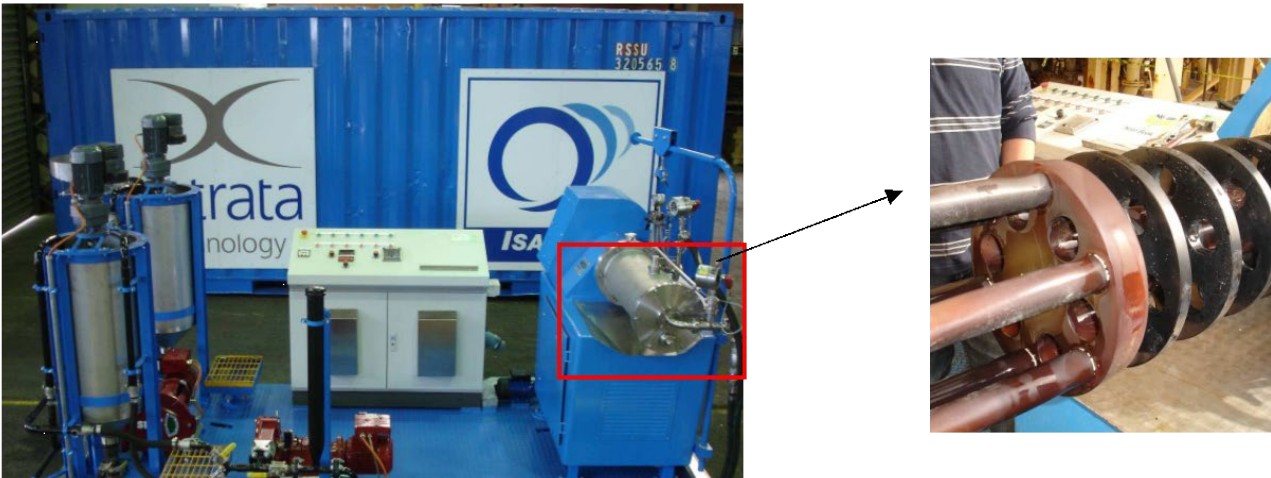

**Figure 4.** Pilot scale M20 and its internal structure (on the **right**).

### 2.2. Sampling of Full-Scale HIGMill and Pilot Tests of IsaMill

The sampling campaigns of both mills were conducted when the steady state conditions were established. For the HIGMill operation, the key performance indicators of the mill were followed for at least 30 mins. once the parameters (i.e., stirred speed and solids content) were changed. In this regard, time-series plots of mill power, flow rate and cyclone pressure parameters were observed. The period where the parameters fluctuated to a minimum extent was accepted as steady state, then the material collection studies were executed to sample mill feed and product streams. A screenshot of the expert system, in which the process data is stored and then plotted, is depicted in Figure 5.

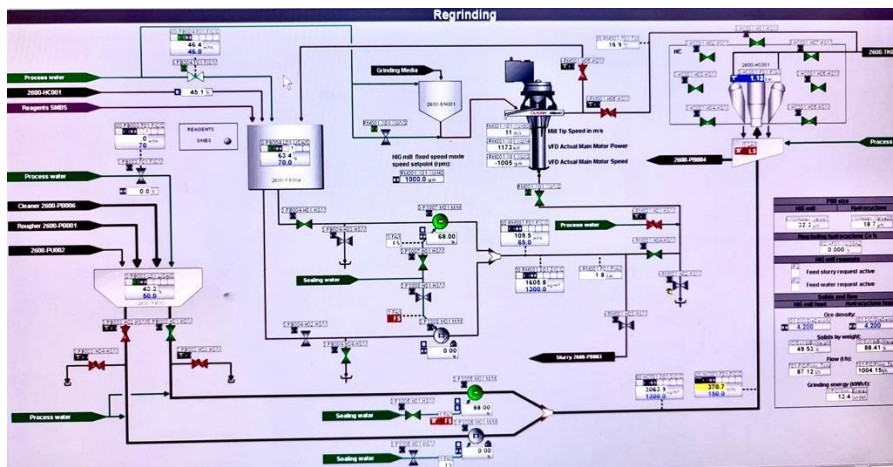

**Figure 5.** Screenshot of the expert system of HIGMill operation.

Pilot scale IsaMill tests were also conducted with the same philosophy. Prior to commencing the sampling campaigns, the power draw of the mill was followed for about 30 min. When the change in power draw was at its minimum, then the system was named as steady state; hence, the samples were collected from the mill feed and discharge. In contrast to HIGMill surveys, IsaMill was operated in a closed loop, meaning that the product is recirculated back to the feed end. In brief, the product of one survey is to be the feed of the upcoming pass. Within the scope, 6 passes were completed.

The HIGMill and IsaMill tests were performed at milling environment, as summarized in Table 2.

**Table 2.** The milling environment of HIGMill and IsaMill.

|  | **HIGMill** | **IsaMill** |
|---|---|---|
| Agitator configuration | 10 castellated + 6 non-castellated | 6 discs |
| Specific gravity of grinding media | 3.7 | 3.7 |
| Bead size (mm) | 3–4 | 2 |
| Bead filling (%) | 64 | 70 |

HIGMill survey was held at varied solids content and tip speed. IsaMill tests, on the other hand, were conducted at the same solids content (≈50%) and tip speed (12 m/s); the feed sample was passed through the mill chamber for 6 times. The complete test matrices are given in Table 3.

**Table 3.** Operating conditions of HIGMill (HM) and IsaMill (IM) surveys.

|  | **40% Solid** | **50% Solid** |
|---|---|---|
| 12 m/s |  | IM Pass 1 to Pass 6 |
| 10.4 m/s | HM T1 | HM T4 |
| 8.3 m/s | HM T2 | HM T5 |
| 6.2 m/s | HM T3 | HM T6 |

*2.3. Material Characterization*

Characterization studies such as determining the particle size, density, work index (Bond test), chemical composition, mineral contents and breakage characteristics were conducted on the samples.

Size distributions were determined from the top size to 8 μm. In this regard, wet sieving (from top size to 38 μm) and cyclosizer techniques (between 38 μm and 8 μm) were utilized. Cyclosizer test apparatus (Figure 6) is comprised of a series of cyclones in which

the flow of water differs. Depending on that, different size classes are collected from each of the glasses.

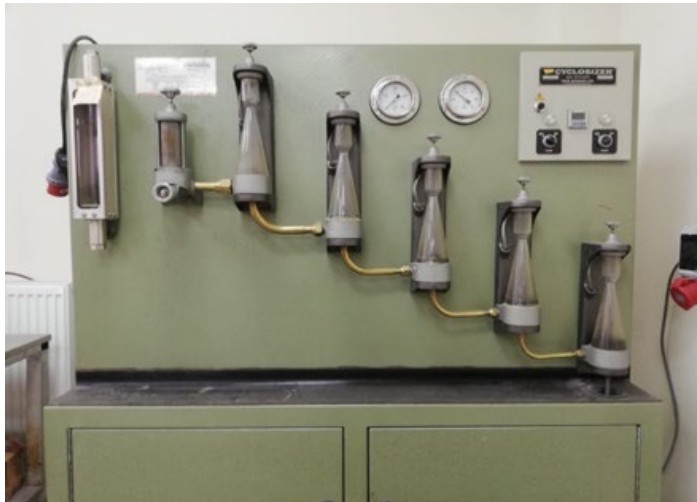

**Figure 6.** Cyclosizer test rig.

Specific gravity measurements were performed with a pycnometer instrument in triplicate. The device is a pear-shaped bottle in which the solid to be measured is placed.

XRF analyses were undertaken to complete the element and mineral information. The test device is depicted in Figure 7.

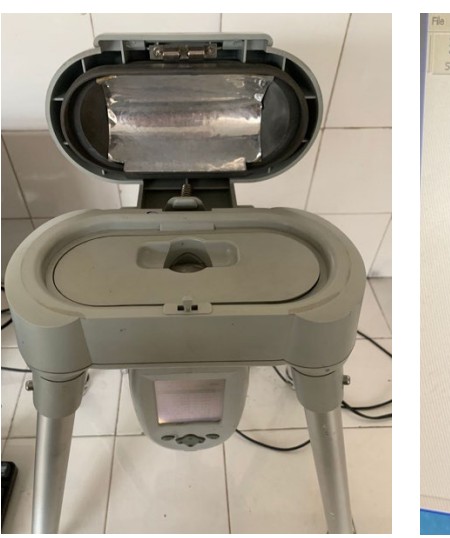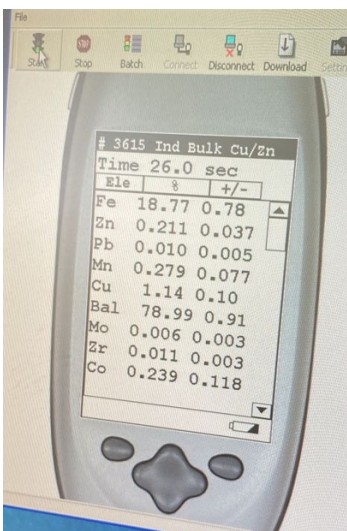

**Figure 7.** XRF test rig (Niton XLt) and its summary display (on the **right**).

Bond work indices of the ores were determined to reveal the variations in the grindabilities. In this respect, standard Bond ball mill with (30.5 × 30.5) cm dimensions were utilized in which the ball size distribution suggested by Bond [20] was charged. It is a lock cycle test, and the procedure is prescribed by Bond [20].

The breakage characteristics of the samples were determined via impact bed breakage tests, which was conducted via drop-weight tester (Figure 8). The technical specifications of the drop-weight tester are summarized in Table 4. The test device includes a steel anvil, a head with a certain weight and an electromagnet, which is used to drop the weight from a height that the potential energy required. The material to be broken is placed in a bed with 3.5 cm diameter and 1 cm of height (Figure 8), then put onto the center of the anvil. Afterwards, the head is dropped, and the samples were collected accordingly.

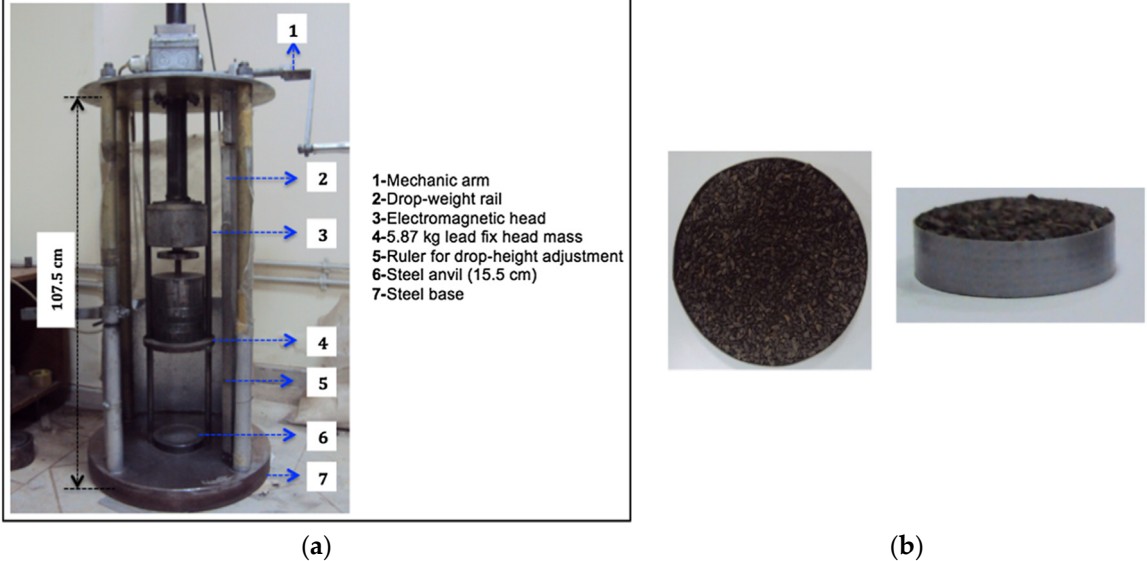

(a)                    (b)

**Figure 8.** Drop weight test apparatus (**a**) and a bed sample prepared for the analyses (**b**).

**Table 4.** Technical specifications of the drop-weight tester.

| | |
|---|---|
| Weight of head (kg) | 5.87 |
| Max. available weight (kg) | 50 |
| Max. height of drop (cm) | 51.5 |
| Diameter of head (cm) | 15.5 |

Evaluating the repeatability of the test apparatus is of prime importance prior to the stage of data interpretation. Such analyses were performed by Eksi [21] at different size intervals and energy levels. The results of the statistical assessments performed at $(-212 + 150)$ μm and at 1 kWh/t of specific energy concluded that there exist no significant difference in the product size distributions after the breakage. Consequently, the test apparatus is said to produce repeatable results.

Table 5 tabulates the details of the drop-weight test works. Within the scope of the tests, 3 size intervals were broken at 3 energy levels.

**Table 5.** Breakage characterization test procedure via drop-weight tester.

| | 0.8 kWh/t | 1.1 kWh/t | 1.7 kWh/t |
|---|---|---|---|
| | **Weight of Sample/Drop Height** | | |
| $(-425 + 75)$ μm | 32.0/21.2 | 32.2/28.9 | 32.1/44.0 |
| $(-75 + 53)$ μm | 31.5/20.8 | 31.5/28.3 | 31.5/43.2 |
| $(-53 + 38)$ μm | 32.5/21.5 | 32.5/29.2 | 32.5/44.5 |

Following the test works, the size distributions of the broken products were measured and then a size-dependent breakage model was utilized to plot t-family curve, which considers the specific energy and particles size (Equation (1)). In this model, $t_2$, $t_4$ and $t_{10}$ values represent the per cent passing from 1/2, 1/4 and 1/10 of the original mean particle size [21–23].

$$t_n = A \times \left(1 - e^{-b.E_{CS}.X}\right) \quad (1)$$

where;

| | |
|---|---|
| $E_{cs}$ | : Specific comminution energy (kWh/t) |
| $A, b$ | : Model parameters |
| $X$ | : Particle size (mm) |

*n*           : 2, 4 and 10

### 2.4. Data Interpretation

The assessments of the milling performances were initially focused on energy-size reduction relationship. In this regard, net energy consumption (Equation (2)) and reduction ratio of the comminution process (Equation (3)) were calculated.

$$Net\ specific\ energy\ (kWh/t) = \left( \frac{Gross\ power - idle\ power}{production\ rate} \right) \tag{2}$$

$$Reduction\ ratio = \left( \frac{F_{80}}{P_{80}} \right) \tag{3}$$

Moreover, the slope of the size distribution curves was calculated via RRSB equation (Equation (4)), which provided valuable information expressing the feed size characteristics. In this regard, n parameter was back-calculated via a non-linear regression technique. The higher the value n, the narrower the size distribution and vice versa.

$$Y = 1 - exp\left[ -\left( \frac{X}{d_0} \right)^n \right] \tag{4}$$

where;

$Y$      : Cumulative passing (%)
$d0$    : Position parameter (size passing from 63.2%, μm)
$n$      : Slope
$X$     : Particle size (μm)

For the breakage comparison of the two ore types, the simulation study was conducted. In this regard, a perfect mixing approach [24] was utilized and the product size distributions were calculated for the same feed size distribution and r/d values. The mathematical expression of the perfect mixing approach is given in Equation (5).

$$f_i + \sum_{j=1}^{i} a_{ij} p_j \frac{r_j}{dj} - p_i \frac{r_i}{d_i} - p_i = 0 \tag{5}$$

where:

$f_i$  : feed rate of size fraction *i* (t/h);
$p_i$  : product flow of size fraction *i* (t/h);
$a_{ij}$ : the mass fraction of particles of size *j* that appears at size *i* after primary breakage;
$r_i$  : the rate of material breakage for particle size *i*;
$d_i$  : discharge rate for particle size *i*.

Within the study, the differences in the stress energies of the technologies were also discussed. This concept was proposed and then used in the scaling up of the stirred mill technology [25]. Kwade and Stender [25] conducted the calculations at different mill geometries to make robust conclusions. The stress–energy equation [26] is given in Equation (6).

$$SE \approx v_t^2 * d_{GM}^3 * \rho_{GM} \tag{6}$$

where:

$v_t$      : tip speed (m/s);
$d_{GM}$   : diameter of grinding media (m);
$\rho_{GM}$   : density of grinding media (kg/m$^3$).

## 3. Results and Discussions

### 3.1. Comparison of the Feed Characteristics

As summarized previously, the two technologies were tested at different circuits. Consequently, the differences in the feed characteristics should be revealed prior to the data processing. In this regard, the size distributions and its shape (n parameter of Equation (4)), chemical assays, specific gravities and the mineral contents were considered. Figure 9 illustrates the feed size distributions to the stirred milling technologies.

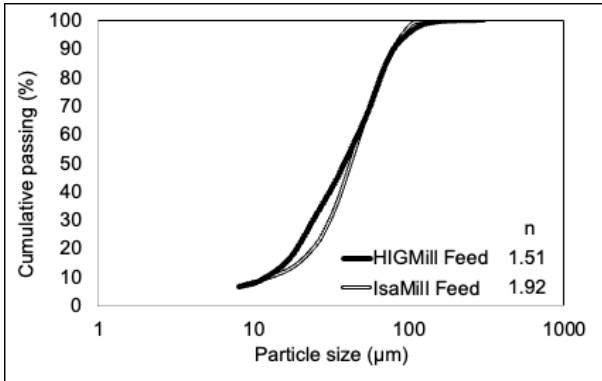

**Figure 9.** Feed size distributions of the two mills.

It is obvious that the $d_{80}$s are very similar, and the shape of the distribution varies considerably. The feed to the IsaMill is steeper than the HIGMill operation. Such a difference in the shape parameter will have impacts on the comminution results as will be discussed in the following sections.

Specific gravity, chemical composition and mineral distributions of the feed samples are tabulated in Table 6. It can be stated that the ores of the comminution circuits have similar characteristics.

**Table 6.** Specific gravities (SG), head assays and mineral distributions of the circuit feed sample.

|  | SG | Cu% | Fe% | Zn% | Chalcopyrite % | Pyrite % | Sphalerite % | Quartz % | Rest % |
|---|---|---|---|---|---|---|---|---|---|
| HIGMill circuit | 3.45 | 2.30 | 29.57 | 0.38 | 6.64 | 57.35 | 0.59 | 24.30 | 11.11 |
| IsaMill circuit | 3.70 | 2.14 | 30.24 | 0.33 | 6.18 | 59.05 | 0.52 | 21.50 | 12.75 |

Grinding and breakage characteristics of the feed samples are summarized in Table 7.

**Table 7.** Work indices and breakage parameters of the circuit feed samples.

|  | Wi (kWh/t) | A | b | Axb | R2 |
|---|---|---|---|---|---|
| HIGMill circuit | 12.80 | 12.17 | 3.25 | 39.55 | 0.97 |
| IsaMill circuit | 12.00 | 22.11 | 1.64 | 36.26 | 0.98 |

The work indices of both ore types are close to each other and both of them are in a medium class according to Napier Munn et al. [22]. Following the drop-weight tests, the breakage characteristics of the ores were investigated by fitting the results to Equation (1). Correlation coefficients, $R^2$, indicate that the fit results are acceptable. As summarized in Table 7, Axb parameter of the two ores showed a small variation whether classified as hard or medium-hard ore types according to Napier Munn et al. [22]. This difference can be attributed to the varied mineralogical compositions of the ores summarized in Table 5. Such conclusions were also drawn by researchers who investigated coupling of comminution parameters with the mineralogical composition [27]. In that study, correlation between the mineral content and hardness parameters were developed.

Figure 10 illustrates the relationship between specific comminution energy and $t_{10}$ parameter. The results show that the two feeds have similar breakage functions, and a small variation was observed at a higher level of comminution energy.

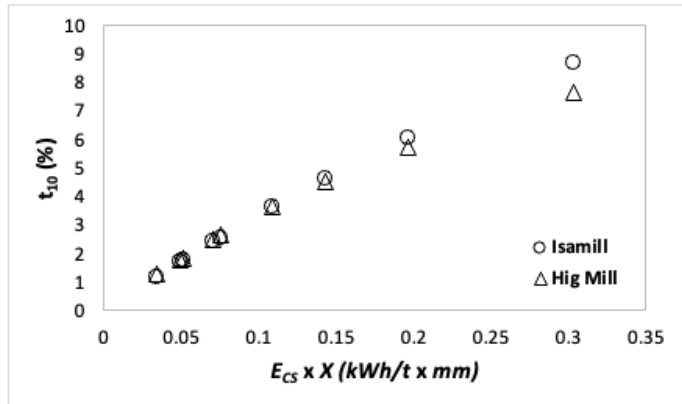

**Figure 10.** Size-dependent breakage model fit results.

The impact of that small variation in breakage function was tested via simulating the size distribution of the mill product (Equation (5)). In this regard, the same rate of breakage and feed size distribution was inputted and only the breakage function was changed from one another (Table 8).

**Table 8.** The variation in the product size distributions of different breakage functions given in Figure 10.

| | Cumulative Passing (%) | |
|---|---|---|
| Particle Size (μm) | P1 | P2 |
| 75 | 100.00 | 100.00 |
| 53 | 97.48 | 98.31 |
| 38 | 94.97 | 94.00 |
| 33 | 93.65 | 92.57 |
| 25 | 88.42 | 86.17 |
| 17 | 70.19 | 68.32 |
| 11 | 46.47 | 49.24 |
| 8 | 40.59 | 39.62 |

The results imply that the product sizes are quite similar to each other. As a result of the assessments, the ores are said to have similar physical properties; hence, the comparison was held at almost similar conditions.

*3.2. Comparison of the Signature Plots*

The performance comparison of the two machines was made with the data collected from the industry (for HIGmill) and pilot scale test works (for IsaMill). For such an evaluation, the result of the pilot scale IsaMill test is expected to predict the performance of the full-scale operation. Thus far, successful application of 1:1 scale-up procedure of IsaMill has been reported in the literature [28–33]. Therefore, the two cases are said to be comparable. Figure 11 depicts the signature plots of the two technologies.

Figure 11 should be evaluated with two aspects, HIGMill data internally and its comparison with the IsaMill. Figure 11 compares the $d_{80}$ values of the product size distributions as a function of the operating conditions, which affects the net specific energy consumption of the milling. There exist two major parameters of HIGMill surveys, which are tip speed and solids content. Within the stirred media milling, increasing the tip speed increases the energy consumption; thus, finer grind is achieved. This phenomenon was also observed for the HIGMill survey (Figure 11). Solids content is the other parameter and the results

show that HIGMill operation at 50% solids is more efficient than 40% solids content as finer size product is obtained for the same energy utilization. Furthermore, the shift in the plots is almost parallel to each other. IsaMill results lie in between the data set of HIGMill. It should be emphasized that, at similar solid content of feed, IsaMill results get close to that of HIGMill towards the finer production. Figure 11 considers reduction ratio parameter as a function of energy consumption since the feed size may vary from survey to survey and between the two technologies. The similar conclusions are also valid. Another observation is on the slope of the trends. Although the HIGMill results shift parallel, the slope of the trend differs for IsaMill. This shows that the two milling technologies have varied performances or responses at coarser and finer production.

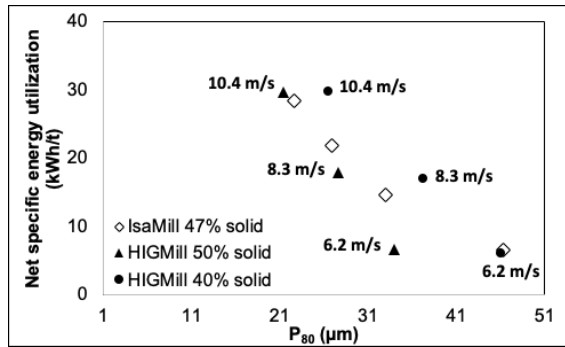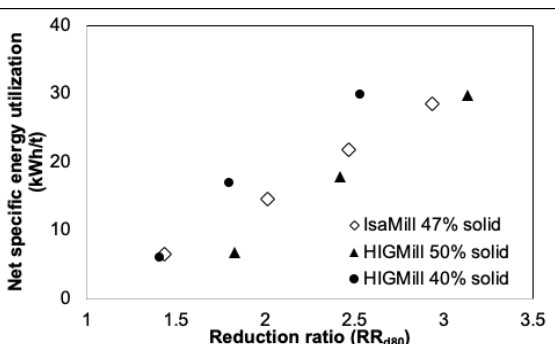

**Figure 11.** Signature plots of IsaMill and HIGMill.

IsaMill operation also has a response to solid content of the milling environment. Jankovic [3] reported that the performance of the mill improves when operated at a denser condition, which is similar to HIGMill behavior. Therefore, IsaMill with 50% solid feeding may give a result similar to that of the HIGMill, which may explain why the IsaMill performance was in between the HIGMill results (Figure 8b).

### 3.3. Comparison of the Product Size Distributions and Energy Values

Figure 12 depicts the differences in the shape of the distribution of the products and the energy utilization of the milling technologies for the same grind size. It should again be noted that the two technologies were operated at different solid contents, which has impacts on the milling efficiency. Consequently, a solid conclusion regarding the energy figures may not be adequate, however necessary. It is thought that the results indicate some important points that should be discussed in detail. For the HIGMill, the product sizes of 35 μm, 27 μm and 20 μm were obtained at 50% solids content and at the tip speeds of 6.2 m/s, 8.3 m/s and 10.4 m/s, respectively. In brief, the IsaMill results were compared at the optimal solids content of HIGMill.

Regarding to the shape of the distribution, the slope values of the two distributions are noticeably different at the production at 35 μm. This can be attributed to already narrower distribution of IsaMill feed compared to that of HIGMill (Figure 9). The narrower distribution at the starting point resulted in a narrower sized product distribution. For finer size productions, $d_{80}$ of 27 and 20 μm, the slopes are close to each other. Although the product of IsaMill is still narrower, it is due to the feed characteristics as discussed previously. Furthermore, the variations in the slope values are not as much as that of 35 μm production. It can be said that the variations in the distributions almost disappear towards the fine particle production.

Regarding the energy consumptions, it is seen that the vertical orientation is more energy efficient than the horizontal milling at coarser size production (Figure 12a). For this case, the energy utilization of HIGMill is 54% less than IsaMill. This difference starts to disappear towards the fine size production. For the grind size of 27 μm (Figure 12b), the energy difference is calculated as 18%, which is noticeably less than 35 μm production, and is still in favor of HIGMill technology. For the finest grind size (20 μm), the energy

assessments changed considerably. For this case (Figure 12c), IsaMill consumes 4.5% less energy than HIGMill technology.

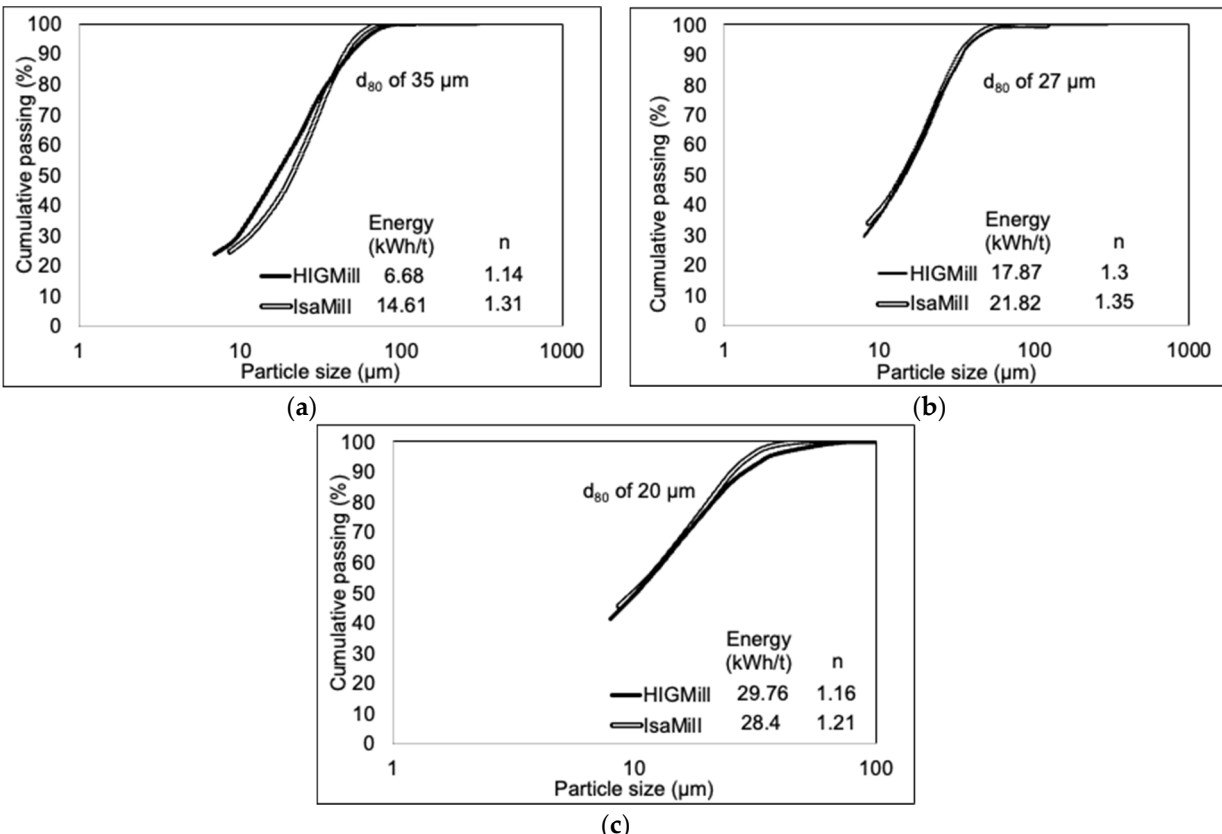

**Figure 12.** The variations in the shape of the product size distributions of (**a**) 35 μm, (**b**) 27 μm and (**c**) 20 μm.

Thus far, the literature has reported the outcomes of DEM and CFD studies on stirred media mills, giving insight on the media trajectories, heat dissipation and energy transfer phenomena. Discussing the literature jointly with the results of this study may help to explain some of the phenomenon better.

More efficient milling of the vertical orientation at coarser size as well as the efficiency of the horizontal orientation at finer size may be explained with the media movement, bead packing and energy absorption of the media. Sinnott et al. [34] in their DEM study investigated the pin mill. In their calculation, it was shown that the deeper the mill, the more intense the high velocity region is. In another study [35], the rate of energy absorption of the beads was found to be higher with the depth. The horizontal orientation on the other hand was found to have more packed beads at the lower half of the mill. This results in less shearing of the beads by the disc [36]. In the meantime, when the vertical slice of the chamber is investigated, it was shown that there exists significant voidage in the upper half of the mill due to the packing of the beads as a result of the gravity meaning that there might be an increased probability of the particles bypassing to the discharge end directly [37,38]. Higher energy utilization of the vertical orientation at the feed end and higher voidage inside the chamber of the horizontal orientation explain the efficiency of the vertical orientation at coarser production.

Sinnott et al. [26] further discussed the comminution action by adding the feeding mode and particles' energy absorption. They concluded for the vertical orientation that feed particles are subjected to a high energy zone at the bottom of the chamber (where the feeding is) and the magnitude decreases towards the discharge end. Therefore, upper volume of the mill may not be utilized efficiently as this zone has relatively less energy input.

However, for efficient fine grinding action, this volume is also required. The horizontal configuration, on the other hand, has a uniformly distributed bead filling along the mill axis; therefore, intense shearing still has to be provided towards the discharge end [35]. As a result, the reason for the performance decrease of vertical orientation at fine particle size can be attributed to these phenomena.

### 3.4. Stress Energies of the Two Mills

Stress energy evaluations give valuable information regarding the optimal condition of the milling. It can also be utilized to compare the efficiency of the stirred milling technologies as reported by Jankovic [3], who compared Tower Mill and SMD. However, for a more robust comparison of the orientations, the milling environment regarding bead size, bead filling, operating feed size, etc., should be aligned. Therefore, this study is expected to meet these criteria as the two technologies had similar operating conditions. Figure 13 depicts the results of the comparison taking the stress energy parameter into account.

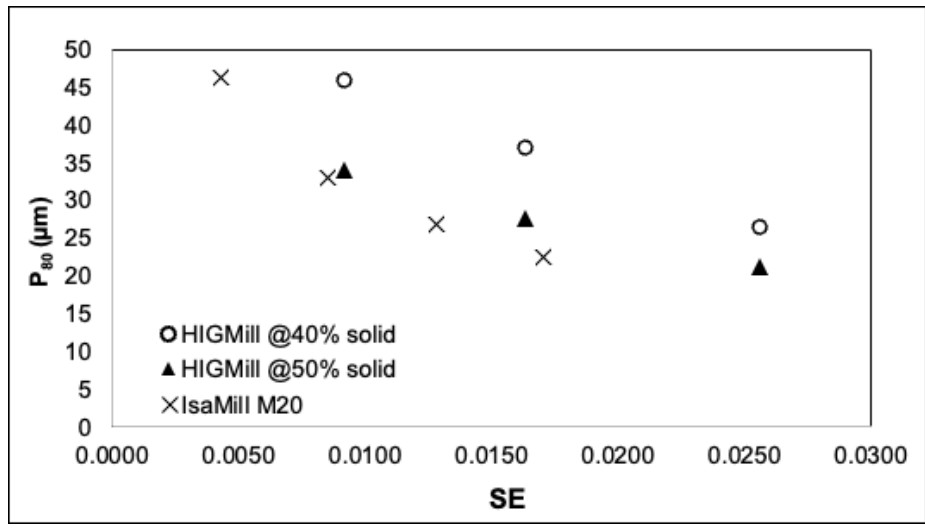

**Figure 13.** Stress energy comparison of IsaMill and HIGMill.

The results show that the two mills have different trends. When the same grind size is considered, the HIGMill operation was found to have a higher stress energy level than that of the IsaMill technology. In other words, the IsaMill operation produced finer product than HIGMill process when the mills are operated at the same stress level. The dominant factor behind these outcomes could be the technological differences. Similar conclusions were also drawn by Cayirli and Gokcen [19], who performed the grinding tests at two different orientations of the same sized stirred mill. They concluded that finer product size was obtained with the horizontal configuration, which was operated at the same stress energy level. Some of the studies utilized DEM technique to compare the energy losses within the mill as a result of the collision events. In this sense, Sinnott et al. [34] and Cleary et al. [35] calculated energy losses at different directions, i.e., shear and normal, for vertical and horizontal orientation, respectively. In their assessments, they considered the modal peaks of the energy spectra. When the two analyses are compiled, it is seen that the horizontal oriented mill is found to have less total energy loss compared to that of the vertical configuration. Hence, a more efficient milling environment may be expected for the horizontal orientation. This may also support the finding of this research. As a result, the outcome of this research is said to coincide with the literature that considered both experimental and mathematical approaches.

## 4. Conclusions

This study aimed at comparing the impacts of the chamber orientation of the stirred milling on the comminution results. Although this topic has been elaborated by some of the studies (Tower mill vs IsaMill, IsaMill vs SMD), there is still a gap in knowledge about the comparison based on the similar milling environment. Therefore, HIGMill and IsaMill technologies' operation fits well for this purpose. Moreover, its novelty lies in being the comparison study on an industrial scale.

The performances of both mills were compared for the ores processed at different mine sites with similar geological characteristics. In order to reveal the variations in the ore characteristics, breakage function, work index and chemical compositions were evaluated. All these results concluded that the two ore types had similar physical properties except the breakage function, which varies slightly in favor of the IsaMill circuit.

The comparisons were evaluated by considering signature plot, particle size distribution and energy utilization as well as the stress energy parameters. Comparison of the signature plots showed that the results of IsaMill results lie in between the data set of HIGMill attributing to the operating solid content or the orientation of the mill chamber. Another fact was the slope of the energy-size reduction trends, which was found to differ for HIGMill and IsaMill. It is obvious that the technologies are different and behave in a different manner for coarse and fine tail of comminution.

Comparison of the energy utilization and the shape of the product size distribution further supports the previous findings. For these evaluations, the mill performances were compared for the same grind size. At coarser production (35 μm), the energy efficiency of the vertical arrangement was higher than the horizontal configuration (54%), which could be attributed to the effects of bead packing, feeding to the mill as well as energy density within the mill chamber. In the mid-zone, the two mills had similar size distribution; however, the energy utilization varied slightly in favor of HIGMill (18%). In the fine grind size (20 μm), horizontal orientation (IsaMill) was found to be more energy efficient. All these results are due to the impacts of the mill orientation. DEM studies in literature showed that the feed particles are subjected to a high energy zone at the bottom of the vertical oriented mill chamber (where the feeding is) and the magnitude decreases towards the discharge end. It explains the higher efficiency of vertical orientation at coarser grind size and reduced efficiency towards the finer production.

Finally, the mill performances were compared according to the stress energy calculations. The plots illustrate that the two orientations had similar results at the coarser end; however, a noticeable difference was observed towards the fine grind size. At the finer end, IsaMill was found to produce finer product when it was operated at the same stress level of HIGMill. This outcome coincides with the literature on both experimental and mathematical approaches.

The outcomes of this study are expected to contribute to the minerals industry, which is required to select an energy-efficient stirred milling technology for a given application. With its detailed discussions, the paper provides an insight into how the energy utilization of vertical and horizontal orientations differs at different grind sizes. Such knowledge is of prime importance for the equipment selection phase of the existing or greenfield projects, which affects the economy of the operation directly.

**Author Contributions:** Investigation, conceptualization, review and editing—M.C.; investigation, conceptualization, methodology, original draft preparation—O.A. All authors have read and agreed to the published version of the manuscript.

**Funding:** This research received no external funding.

**Data Availability Statement:** Data can be shared upon a request.

**Acknowledgments:** The authors would like to thank Tolga Sert, Cumhur Erdem Karahan, Özgün Darılmaz, Nurettin Alper Toprak for the contributions in the sampling surveys.

**Conflicts of Interest:** The authors declare no conflict of interest.

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
