# Peer review of "Performance Comparison of the Vertical and Horizontal Oriented Stirred Mill: Pilot Scale IsaMill vs. Full-Scale HIGMill"

_minerals, doi:10.3390/min13030315_

Round 1
Reviewer 1 Report
In this paper, some valuable conclusions are obtained by comparing the performance of two different types of mills. However, there are still several questions to be answered by the authors.
1) In lines 123-135, please supplement the model and parameters of the experimental equipment.
2) In line 199, Table 20 should be replaced by Table 9. In addition, the product A*b value of crushing parameters A and b can represent the hardness of minerals. A*b value less than 30 is very hard, 30-38 is hard, and 38-43 is medium hard. However, from the results in Table 9, the A*b value of IsaMill is 36.26, which belongs to hard level, and the A*b value of HIGMill is 39.55, which belongs to medium hard level. Because the hardness of minerals largely determines the difficulty of grinding, I would like to ask whether the authors have done repeated experiments and explain why the hardness of the two minerals is different.
3) In line 216-218, will there be any error when Isa Mill is scaled up 1:1? Are there any more references as support?
4) In lines 267-289, can the authors add more literature to support your view?
5) This paper is about industrial application, so please briefly describe how the research results of this paper will help practical application?
Reviewer 2 Report
The authors conducted research related to two steer media mills – IsaMill and HIGMill. They compared the performance of these mills through several approaches and supported the obtained results with literature sources. However, there are issues given in the manuscript that remain somewhat unclear and should be improved. Remarks follow below.
1. Chapter 2.2: Give in more detail how the test procedure in the IsaMill was carried out (number of passes and similar) and how the indicators of the grinding and grinding product were measured. In addition, provide a more detailed description of how the HIGMill grinding indicator were measured so that the results from both procedures are comparable.
2. Table 3 is somewhat confusing, it should contain numerical values instead of check marks and it should give the operating conditions of both mills in parallel.
3. Give one of the literature sources for Equation 1. After the parameter n (Line 145), give the definition and unit for Ecs.
4. Table 4 is also confusing at first glance and should contain numerical values instead of check marks. It can be supplemented with parameters such as 'weight of sample' and 'drop height'.
5. Line 157: change m to n.
6. Lines 193 do 201: To reduce the number of small tables, mineralogical composition and chemical composition can be given through one table. Also, the parameters that characterize the comminution can be given through one table. Add value Axb to this table. In the main text, give the meaning of parameter R2.
7. Line 207: Equation 6 or Equation 5?
8. Section 3.2: Grinding experiments in the HIGMill were performed in industrial conditions at different tip speeds of impeller, but this is not explicitly seen from the Figure 8. Besides solid content, this phenomenon, i.e. the influence of tip speed should also be discussed.
9. Section 3.3: Under what conditions were obtained the particle size distribution curves of the HIGMill outputs (solid phase content, impeller speed) shown in Figure 9? Discussion in the text should be performed.
10. Lines: 250: HIGMill or IsaMill?
11. In my opinion, the discussion with literature supporting the results and the conclusion are written quite well.
Reviewer 3 Report
The study is of a review and research nature. Two types of ore grinding mills were extensively characterized, taking into account the energy benefits in favor of one of them. The correct process of selecting experimental studies and exhaustive analysis of the results. I have no comments
Author Response
Authors would like to thank to the Reviewr for valuable and supportive contribution.
Round 2
Reviewer 2 Report
Thanks to the authors for the corrections made and the improved quality of the manuscript.